# Predicting Fitness Centre Dropout

**DOI:** 10.3390/ijerph181910465

**Published:** 2021-10-05

**Authors:** Pedro Sobreiro, Pedro Guedes-Carvalho, Abel Santos, Paulo Pinheiro, Celina Gonçalves

**Affiliations:** 1Sport Sciences School of Rio Maior (ESDRM), Polytechnic Institute of Santarém, 2001-904 Santarém, Portugal; abel-santos@esdrm.ipsantarem.pt; 2Life Quality Research Center, 2001-904 Santarém, Portugal; 3Research Center in Sports Sciences, Health Sciences and Human Development (CIDESD), 5001-801 Vila Real, Portugal; pedro.guedes.carvalho@gmail.com (P.G.-C.); celinag@ismai.pt (C.G.); 4Departamento de Ciências e Tecnologia, Universidade Aberta, 1269-001 Lisbon, Portugal; paulo.pinheiro@cedis.pt; 5Sport Department, University of Maia, 4475-690 Maia, Portugal

**Keywords:** sports management, sports services, fitness, machine learning algorithm, dropout prediction, gradient boost classifier

## Abstract

The phenomenon of dropout is often found among customers of sports services. In this study we intend to evaluate the performance of machine learning algorithms in predicting dropout using available data about their historic use of facilities. The data relating to a sample of 5209 members was taken from a Portuguese fitness centre and included the variables registration data, payments and frequency, age, sex, non-attendance days, amount billed, average weekly visits, total number of visits, visits hired per week, number of registration renewals, number of members referrals, total monthly registrations, and total member enrolment time, which may be indicative of members’ commitment. Whilst the Gradient Boosting Classifier had the best performance in predicting dropout (sensitivity = 0.986), the Random Forest Classifier was the best at predicting non-dropout (specificity = 0.790); the overall performance of the Gradient Boosting Classifier was superior to the Random Forest Classifier (accuracy 0.955 against 0.920). The most relevant variables predicting dropout were “non-attendance days”, “total length of stay”, and “total amount billed”. The use of decision trees provides information that can be readily acted upon to identify member profiles of those at risk of dropout, giving also guidelines for measures and policies to reduce it.

## 1. Introduction

Dropout has always been an ongoing concern for fitness centre managers. According to the International Health, Racquet and Sports Club Association [1], six causes of dropout have been identified: the high number of members in facilities; dissatisfaction with employees; a lack of interest shown by staff; disappointment with programmes and activities provided, and the inaccessibility or lack of response from individuals in charge. Previous studies have already addressed the difficulties faced by fitness centres in retaining members, in an attempt to reduce high rates of dropout [2,3].

The fitness sector is well known for its high incidence of dropout [4], and Portugal is no exception where rates of 65% were experienced in 2018 [5]. Emeterio, García-Unanue, Iglesias-Soler, Felipe, and Gallardo [6] showed that only 30–60% of members continue attendance in the second year at fitness centres. Whilst these studies focused specifically on the nature of the fitness services supplied, Sperandei, Vieira and Reis [7] have stated that the most important issue is to identify the profile of those individuals at the highest risk of dropout.

To achieve such aim, the authors applied a regression model that revealed that age, previous level of physical activity, body mass index and motivation to lose weight, hypertrophy, health and aesthetics, were the main variables related to the risk of dropout. Gender has also been considered relevant, reinforced by the differences found in physical activity between the sex [8,9]. In addition, Pawlowski, Breuer, Wicker and Poupaux [10] have suggested that age and monthly income are also relevant factors to consider in understanding commitment to sports services. Furthermore, the identification of daily routines in the pursuit of a sport is another aspect identified as critical to retention in fitness centres [9,11]. These combined variables influence the likelihood of dropout in fitness centres. Surprisingly, even individuals with the best possible combination of variables have a high risk of giving up before the completion of a 12-month period [7].

Retention and dropout are two sides of the same coin. The literature shows that fitness centre profitability is key to their survival and that member retention is a critical element in achieving such profitability [12,13]. Consequently, increasing member retention rates in fitness centres will reduce the costs of marketing required to attract new members to replace lost ones.

Bodet [14] states that customer retention reflects the intentions of members to renew their membership. Therefore, the prediction of dropout may allow fitness club managers to take countermeasures to mitigate it, given that retaining existing members is more profitable than recruiting new ones [15].

Machine learning can be used as a prediction tool. It is defined as an automated process that extracts patterns from data [16], enabling the anticipation of events that allows the development of counteractions. Machine learning can be used to inform the development of customer retention strategies based on existing data [17]. To our knowledge there are no available studies using this method applied to customers’ available data held by fitness centres. Therefore, it would be sensible to experiment it first, using available data to ‘train’ a model and to make generalisations [18].

Dropout prediction consists of identifying two possible results, either dropout or non-dropout; a set of techniques is used to learn how to model the relationship between a set of descriptive variables and a target variable, using a set of historical examples [16]. The performance evaluation in the prediction uses typically the prediction accuracy (the number of correct predictions against the total number of predictions), which depends on identifying the appropriate algorithm and allocating the data to the training and testing set [19].

The use of decision trees allows us to extract actionable knowledge [20,21]. The existing research related to the prediction of dropout in fitness centres mentions process accuracy [6,22], but there is a lack of research exploring actionable knowledge in the prediction. There is, however, an exception in the study of Pinheiro and Cavique [23], which proposed workflows to develop actions according to customer profiles, thus identifying a gap in the research which we would like to address.

The aim of this research is to predict member dropout in fitness centres using machine learning algorithms, through the identification of member profiles, which can then be used to inform management policies designed to increase the profitability of these centres. After this introduction, this study comprised the following sections: (1) the methodology with dataset, machine learning models used, and performance of the prediction; (2) explanations of the results are then present and (3) discussed in relation to existing studies. Finally, (5) conclusions are stated, and references noted.

## 2. Methods

Based on a fitness centre customer dataset we applied machine learning algorithms to find useful management information to reduce the incidence of potential dropout.

### 2.1. Dataset

In this study, data from 5209 fitness members was analysed (mean age = 27.87, SD = 11.80 years) from a Portuguese fitness centre located in Lisbon. It has an extensive fitness service that includes Group Classes, Cardiovascular Training in the different Exercise Rooms, Personal Trainer Indoor/Outdoor service and an Athletics School. Although it is a club that stands out for its services and different facilities, its typology fits into the typology of a conventional Gym and Health Club, according to the typology of Pedragosa and Cardadeiro [24]. The customers pay a monthly fee for group activities or per sessions with a Personal Trainer. The average number of registrations per month was 9.34 ± 8.22. The data was collected from the software e@sport (Cedis, Portugal) between 1 June 2014 and 31 October 2017. The information retrieved covered the registration data, payments made and frequency of visits. Dropout was defined according the contractual conditions of membership in the fitness centre; it occurred when the member gave notice of an intention to terminate the contract or did not pay the monthly fee within a period of up to 60 days. The data processing involved the following steps: (1) initial dataset (*n* = 5216); (2) removing missing values (*n* = 5210); (3) removing incorrect data (*n* = 5209) and (4) final dataset and analysis results (*n* = 5209). Data anonymity was ensured by removing all personal information before recovering data from the software used by the centre.

The data processing was developed with Anaconda and IPython [25] using Pandas [26] and NumPy [27] software. The data retrieved is presented in Table 1. The dataset includes 12 variables for 3381 males and 1834 females. During the period of the study only 644 remained members; this corresponds to 12.3% of the total. The mean age of the 5215 customers was 27.87 ± 11.80 years. These customers accomplished an average number of visits of 0.89 ± 0.76 per month. The total number of visits to the sports facility was 29.04 ± 41.13. The number of registrations per month was on average 9.34 ± 8.22.

### 2.2. Machine Learning Classification Models

In this study we investigated and compared the performance of several machine learning classification algorithms: Logistic Regression (LR), Decision Tree Classifier (DTC), Random Forest Classifier (RFC), and Gradient Boosting Classifier (GBC). The algorithms were selected according to their availability in the library scikit-learn [28]. These models allow gain insight into the relationships between the dependent variables explaining a target variable [29]. The tree-based models select the variable to split based on the minimisation of the impurity.

Logistic Regression (LR) is a statistical process for estimating the relationship between a dependent variable and one or more predictor variables. It is used to model the probability of a class event (e.g., dropout or non-dropout). LR models the probability of output using a cut-offs value to classify the inputs using a value between 0 and 1 representing the odds of producing one outcome versus another [30].

Decision tree classifiers (DTC) are tree-shaped structures representing sets of decisions to generate classification rules for a dataset [31]. DTC is an algorithm that allows the creation of a model to predict the value of one variable using several dependent variables [32]. The principle of decision trees is to divide the data into smaller datasets based on various descriptive features enabling the identification of sets that fall under one label applying simple decision rules, where the slip decision is based on the increase of purity [29]. Their main advantage is that they are easy to interpret and require little preparation of data (i.e., there is no need to normalise the data) [29].

Random Forest Classifiers (RFC) build a large collection of de-correlated trees, and then average them [33]. The trees are created using subspaces of the feature space, where the features are randomly selected from each subspace [34]. The resulting trees are used to create a classification which reduces the limitations of decision trees [33], given that they are prone to overly adapt to the training data [35]. The simplicity of this approach contributes to its popularity [36].

The Gradient Boosting Classifier (GBC) is an iterative model using several weak learners, mainly decision trees [37]. The models are added sequentially (boosting), combining weak, simple models to obtain a stronger prediction [38]. A weak classifier is one whose error rate is only slightly better than random guessing [39]. The principle is to construct learners in each iteration, reducing the weight of the well-classified, and increasing the weak learners; when the iterations are finished the weighted weak learners produce a strong classifier [35].

### 2.3. Model Performance

The extracted features, with the exception of dropout (the target variable), were used to ‘train’ the model to predict dropout. The prediction was developed using the data not used to ‘train’ the model. To evaluate the performance of the models, we used the holdout method, using a proportion of the data to ‘train’, and a proportion to test. Part of the data (70%) was used to ‘train’ the model, and the other 30% to test the model [40]; this means that 3646 individuals were used to train the model and 1563 to test it, to ensure that the proportion of cases in the train/test stratification was defined in the target variable. The class imbalance was addressed using the balanced parameter to adjust the weights inversely proportional to class frequencies in the input data using the library scikit-learn [28]. The hyperparameters optimisation was developed using grid search targeting AUC as the optimisation goal considering the discriminatory power [22]. The optimisation included solvers, penalty and c_values for Logistic Regression, *n*_estimators, max_features, max_depth for the Gradient Boosting Classifier, and *n*_estimators, max_features, and max_depth as criteria for the Decision Tree Classifier and Random Forest Classifier.

The optimised parameters were applied to the model prediction. The performance calculation was based on the confusion matrix, a contingency table with two dimensions, predicted and actual, with reference to two classes of dropout and non-dropout, comparing the predicted dropout against the real dropout.

The outcome of the binary classifier has the following possibilities: True Positive (TP—‘no dropout’ with a predicted outcome of ‘no dropout’), True Negative (TN—‘Dropout’ with a predicted outcome of ‘dropout’), False Positive (FP—‘No dropout’ with a predicted outcome of ‘dropout’), and False Negative (FN—‘Dropout’ with a prediction of ‘no dropout’). The performance of the model was calculated using the metrics sensitivity, specificity, precision, F1-score and AUC [41]:Sensitivity (SN): True positive rate = TP/(TP + FN);Specificity (SP): True negative rate = TN/(TN + FP);Precision: True predicted ‘no dropouts’ against all predicted ‘no dropouts’ true or not TP/(TP + FP);F1-Score: Combines precision and sensitivity representing their harmonic mean 2 × TP/(2 × TP + FP + FN);Receiver Operating Characteristic (ROC) Curve: Representing the model capability to distinguish dropout and non-dropout. Higher AUC (Area Under the Curve) better model prediction 0 and 1.

## 3. Results

We examined the machine learning models and evaluated their performance. The hyperparameter optimisation using grid search on the training data achieved an area under the curve (AUC) score for LR of 0.845, DTC 0.898, RFC 0.947, and GBC 0.965. The performance of the prediction was calculated by comparing the model prediction against the actual observed values. The accuracy, sensitivity, precision, F1 Score, and AUC are presented in Table 2. The algorithm with the best performance was GBC with an accuracy of 0.955, specificity of 0.968, precision of 0.760, F1 Score of 0.819, and AUC of 0.873. The GBC algorithm was only surpassed by the RFC algorithm regarding AUC (0.890, Table 2), which in our study represents the lower capability to distinguish dropout and non-dropout.

The variable ‘dayswfreq’ was the highest ranked for dropout prediction in the DTC, RFC and GBC machine learning classification models with a value of 54.21% (DTC), 43,29% (RFC) and 35% (GBC). This was followed by ‘months’ (14.62% DTC, 14.27% RFC and 14.71% GBC) and ‘tbilled’ (17.68% DTC, 9.7% RFC and 14.47% GBC). The analysis of partial dependence plots (Figure 1) in the algorithm with higher AUC allowed to understand the direction of the associations. The analysis of the most relevant variables shows that higher values in the ‘dayswfreq’ increases the probability of dropout. Higher values in ‘months’ and ‘tbilled’ decreases the probability of dropout.

The algorithms DTC, RFC and GBC allow the visualisation of the trained model to understand the decision tree used to predict dropout (1 = yes or 0 = no); this makes it possible to identify the variables more clearly, by simplifying the representation of trees to three levels.

In Figure 2, the decision tree used to train the algorithm DTC is shown, showing ‘free_use’ as a first criterion, followed by ‘months’ and days without attendance. The decision tree used in DTC, ‘tbilled’ was the first node used, followed by ‘dayswfreq’ and ‘age’, which reflects the binary decision used in the model. It is possible to extract if-then-else rules to identify a dropout profile related to members that have lower ‘tbilled’ ≤ 365.025, ‘dayswfreq’ > 7.5, and age ≤ 42.5 to represent 1244 members that dropped and 30 that did not. This allows us to identify the rule (‘tbilled’ ≤ 365.025) and (‘dayswfreq’ > 7.5) and (age ≤ 42.5).

## 4. Discussion

This research evaluates the performance of machine learning algorithms to predict the incidence of dropout, using data available for fitness centre members. We created a predictive model from behavioural variables without using questionnaires, which makes a significant contribution to the previous literature, allowing managers to identify the most responsible variables for dropout. The best models RFC and GBC achieved a performance greater than 90% in accuracy, sensitivity, precision and F1 score, surpassed only by DTC with a specificity of 0.819 and precision 0.969. The AUC in RFC and GBC was very similar, respectively, 86.5% and 86%, showing the capability of the algorithm to distinguish between dropout and non-dropout.

The imbalanced dataset was addressed using the higher weights in the minority class (dropout = 0) with 12.3% for the algorithm ‘better learn from the rare class’. GBC had the best performance in overall accuracy, sensitivity to predict dropout and F1 Score. This might mean that it should be an option for predicting which members are likely to dropout. On the other hand, the best results in terms of specificity and precision were achieved using the DTC algorithm. The best performance in the AUC was achieved with RFC, with GBC achieving a similar performance. The good performance of GBC has already been identified in other studies (e.g., [37]) and seems to be the best model to predict dropout.

The best performance algorithm may be used to identify the risk of dropout, using data from its information systems. This enables the development of management countermeasures to reduce dropout. However, the model accuracy should not be the only option to assess dropout. Sensitivity and specificity should be giving insights into the model performance predicting true positives and true negatives if we are interested in predicting the accuracy of dropout or retention. These indicators should trigger actions related to self-motivation and goal theory, in accordance with existing studies [42]. The results give us guidelines about which variables fitness managers should consider in relation to the importance attributed by the algorithms’ performance such as ‘dayswfreq’, ‘months’, ‘tabilled’, ‘nrenewals’, ‘free_use’, and ‘age’ as variables to monitor periodically. The tree-based models select the variable importance based on the minimisation of the impurity of the nodes, which led to a similar result in the variables importance.

In this research, the variable ‘dayswfreq’ was identified as the most reliable variable to predict dropout from the tree models tested, which has also been identified in other similar studies [43,44]. A similar result was also found by Ferrand, Robinson, and Valette-Florence [11], who reported that a regular fitness centre attendance is vital for member retention and positively impacts profitability. These authors have added that the frequency with which members go to the fitness centre is synonymous with satisfaction, and this consequently impacts their decision to stay. Fitness industry managers report that frequency of use is reflected in staying in the centre for longer [45].

The second variable found to be the most reliable in predicting dropout was ‘months’. Garcia-Fernández et al. [46] found that the longevity of their membership in the fitness centre could influence future intentions. Their results showed that membership duration of one to two years better predicts future intentions, specifically when it is concerned with general satisfaction. With reference to this, Surujlal and Dhurup [47] suggest the need to develop strategies to ensure membership renewal plays an essential role in improving customer relationships and retention. In fact, our results also confirmed the importance of the variable ‘months’ in predicting dropout. This could be related to the length of stay giving rise to a lower probability of dropout as identified in other studies [11,14]. Emeterio et al. [22] proved that the length of membership was a statistically significant variable in predicting member dropout. These results provide guidelines for the ‘duration of member registration’ as a variable to follow.

The third variable that we found to predict dropout is ‘tbilled’. In fact, financial aspects were also identified in the literature as a variable to consider [6,22]; however, there are no applied studies including this variable in such a context. This is why we believe that taking into account member’s investment in fitness centre services can help predict their likelihood to dropout.

The use of decision trees has the advantage to support the extraction of actionable information [20,21]. Several studies have tried to predict dropout from fitness centres [6,22] but the use of decision trees was not adopted. Our study contributes to this discussion since the performance of algorithms was analysed that allows the generation of decision trees.

Survival trees may suggest action plans to change a customer’s status from a predicted ‘demotivation’ to ‘retention’. This is useful because it enables us to focus on actions targeted to specific groups which are more susceptible to dropout (e.g., [21]). Pan et al. [21] encourage us to think of the profiles identified in the decision trees most susceptible to dropout, to develop actions to move customers to nodes of the trees more associated with retention. This should be developed with the exception of anything which we cannot change in the customer, e.g., gender or age, by focusing on other variables that managers can influence by their actions, such as ‘dayswfreq’ or ‘months’. Pinheiro and Cavique [23] have also proposed the adoption of workflows according to the existing profiles determined by decision trees.

As in every ‘train’ study there are some limitations. Despite the importance of consulting the variables considered relevant in the existing literature, access to existing databases in fitness centres is not always permitted.

The use of post processing techniques does not identify actions to be developed [21]. The if-then-rules extracted from decision trees (e.g., (‘tbilled’ ≤ 365.025) and (‘dayswfreq’ > 7.5) and (age ≤ 42.5)), can be used to apply specific workflows according to the node of the tree, targeting retention actions to this member profile. The use of the customer lifetime value can also be an indicator to measure the performance of the approach.

The low retention rate (12.3%) identified in the study data should be replicated in other contexts, before inferences are made. The data is restricted to a single fitness centre designed with specificities such as the region where it is located, type of service being provided, and variety of facilities, however, allowed to investigate and demonstrate the suitability of decision tree machine learning algorithms to predict dropout and identify members’ profiles. The analysis of member behaviour data can be undertaken to mitigate member dropout [6]. Segmenting the members based on the likelihood of dropout may be useful to improve the effectiveness of loyalty strategies and to optimise the organisation of human and material resources [6]. Further research should be conducted on the extraction of actionable knowledge from decision trees to increase an objective function such as profit [21].

The algorithm based on GBC should be considered also if we want to predict correctly members that only drop out, and DTC to predict member retention, which could be relevant in considering the management actions taken against the risk of dropout. However, RFC should be analysed also considering its the performance in relation to AUC.

## 5. Conclusions

GBC achieved the best results in the accuracy (95.5%), sensitivity (98.6), and f1 score (97.5). RFC showed a good performance in the AUC (86.5%), but GBC achieved almost the same result (86%). DTC had the best results in specificity (81.9%) and precision (97%). The most important variable in the prediction of dropout was ‘dayswfreq’, achieving more than 50% of the explanation in DTC, 43.29% in RFC and 35% in GBC. This was followed by ‘months’ (14.62% DTC, 14.27% RFC and 14.71% GBC) and ‘tbilled’ (17.68% in DTC, 9.7% in RFC and 14.47% in GBC).

According to our study, sports managers are invited to analyse these variables regularly if they want to improve retention and reduce dropout; furthermore, it also recommends the use of machine learning algorithms based on decision trees to help managers extract actionable knowledge and rules to inform workflows to reduce dropout, using the decision trees and the dropout risk to create if-then-else rules to support the definition of workflows with countermeasures according to member characteristics.

Forthcoming studies should aim at replicating this study in other fitness centres and health clubs, in order to validate the results in other contexts.

## Figures and Tables

**Figure 1 ijerph-18-10465-f001:**
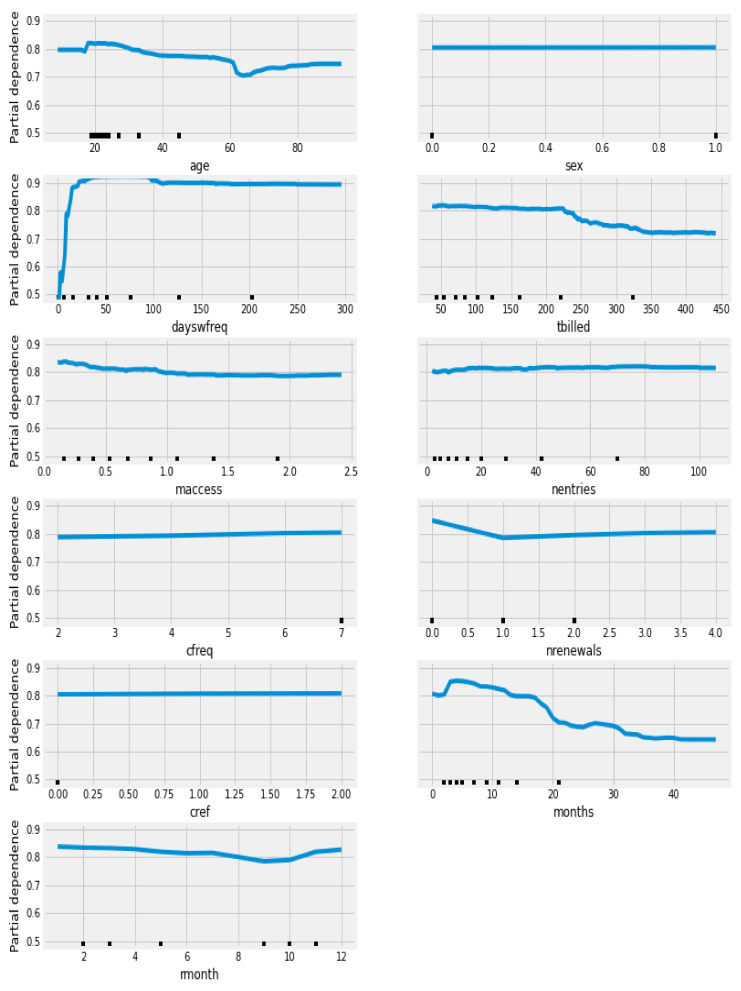
Partial dependence plot of the variables predicting dropout using train-test in the RFC algorithm.

**Figure 2 ijerph-18-10465-f002:**
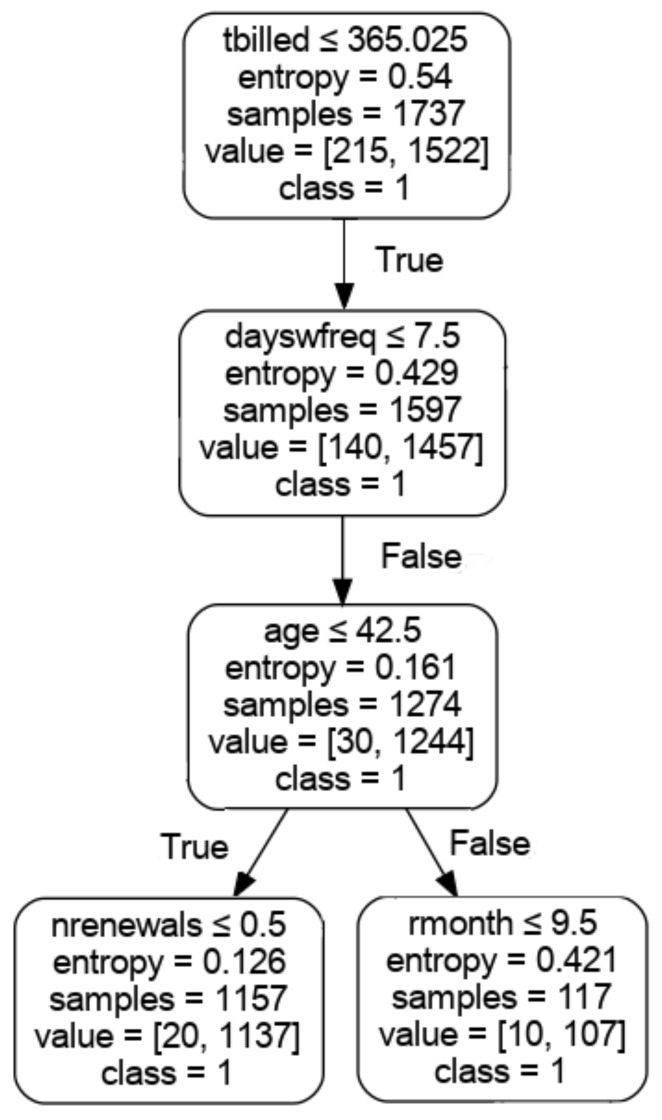
Example of a decision tree created using the DTC algorithm.

**Table 1 ijerph-18-10465-t001:** Variables extracted for each subject of the sample and descriptive statistics.

Variable	Description	Min	Max	Mean (SD *)
age	Age of the participants in years	9	93	27.87 (11.80)
sex	Sex (0—female; 1—male)	0	1	0.35 (0.48)
dayswfreq	Non-attendance days before dropout	0	991	76.40 (101.80)
tbilled	Total amount billed during the registration period (values in euros)	3.60	3747.20	155.32 (162.45)
maccess	Average number of visits per week	0.01	10.33	0.89 (0.76)
nentries	Total number of visits to the fitness centre that the member made during the registration period	1	585	29.06 (41.15)
cfreq	Weekly contracted accesses	2	7	6.86 (0.72)
nrenewals	Number of registration renewals	0	4	0.78 (0.90)
cref	Number of member referrals	0	2	0.01 (0.08)
rmonth	Registration month	1	12	6.72 (3.53)
months	Member enrolment (total time in months)	0	47	9.35 (8.22)
dropout	Measurement of members’ commitment (0 = active, 1 = dropout)	0	1	0.88 (0.33)

* SD—standard deviation.

**Table 2 ijerph-18-10465-t002:** Comparison of the performance of the machine learning classification models using holdout validation.

Performance	LR	DTC	RFC	GBC
Accuracy	0.785	0.839	0.920	0.955
Sensitivity	0.785	0.842	0.938	0.986
Specificity	0.786	0.819	0.790	0.735
Precision	0.963	0.970	0.969	0.963
F1 Score	0.865	0.901	0.953	0.975
AUC	0.786	0.830	0.865	0.860
CI * (Lower, Upper)	(0.759, 0.812)	(0.807, 0.853)	(0.845, 0.884)	(0.840, 0.881)

* AUC 95% Confidence Interval (CI).

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
