# Peer review of "Predicting Fitness Centre Dropout"

_ijerph, 2021, doi:10.3390/ijerph181910465_

Round 1

Reviewer 1 Report

The manuscript aims to study the dropout reasons of fitness customers in Portugal. Authors analyze behavioral variables without using questionnaires. This topic seems relevant and original. However, some aspects need revision. 

INTRODUCTION

The background and state of art are well developed. As a minor issue, I suggest authors unify some contents and avoid one-sentence paragraphs.

METHOD
Despite you have used a large amount of data (5k) and the regression model seems to be well developed, I have a serious concern about you have developed the research in a single fitness center. Authors must provide more information about the fitness center (services provided, number of members, geographical situation, etc.) and especially its economic status and prices (low-cost, mid-market, or premium). This info can affect seriously the results, and maybe this study should be re-named as a case study of a fitness center in a specific category.

Minor review: please remove the numbers of lines 179 and 180. In addition, the size of figure 2 is not appropriate.  

DISCUSSION 

The authors have to improve the limitations paragraph with the previous comments about the characteristics of the fitness center and if these could affect the results.

Author Response

We would like to thank the reviewer for his/her interest and time in assessing our manuscript. It was of great appreciation to receive the thoughtful comments and suggestions to improve the manuscript made by the reviewer. Based on them we revised our manuscript in order to improve its content and quality.

Below we encompass the responses to the raised issues.

If there are any further changes or clarifications needed, please let us know. We will be happy to address them.

Comment: I suggest authors unify some contents and avoid one-sentence paragraphs (Introduction).

Answer: Thank you for your positive suggestion. Some contents were unified, e.g. term consumer or customer was changed to member. The one-sentence paragraph at line 31was joined with the previous one. The same change in the paragraph that started at line 50.

Comment: Despite you have used a large amount of data (5k) and the regression model seems to be well developed, I have a serious concern about you have developed the research in a single fitness center. Authors must provide more information about the fitness center (services provided, number of members, geographical situation, etc.) and especially its economic status and prices (low-cost, mid-market, or premium). This info can affect seriously the results, and maybe this study should be re-named as a case study of a fitness center in a specific category.

Answer: We thank the reviewer for pointing this out, was added information about the fitness center. It has an extensive Fitness service that includes Group Classes, Cardiovascular Training in the different Exercise Rooms, Personal Trainer Indoor/Outdoor service and an Athletics School. Although it is a club that stands out for its services and different facilities, its typology fits into the typology of a conventional Gym and Health Club, according to the typology of Pedragosa and Cardadeiro [24]. The customers pay a monthly fee for group activities or per session with a Personal Trainer.  The number of customers is 5209 members, which we provided at the beginning of the paragraph should. Regarding this, we are not sure about the number of members, or we misunderstand this. This was added to line 95.

Comment: Minor review: please remove the numbers of lines 179 and 180. In addition, the size of figure 2 is not appropriate.

Answer: The lines 179 and 180 were removed. The size of figure 2 was adjusted and represented part of the decision tree to provide an easier interpretation.

Comment: Discussion: The authors have to improve the limitations paragraph with the previous comments about the characteristics of the fitness center and if these could affect the results.

Answer: The reviewer was right in his/her suggestion, all authors agreed, and the discussion. The change was added to line 310.

Reviewer 2 Report

Firstly, I would like to appreciate the opportunity of reviewing this article. I believe it is a very interesting paper with great implications for the future of the fitness industry. I would also like to congratulate the authors for the great and rigorous work performed. The scientific rigorousness is clear and it is somehow challenging for a reviewer to write a review of a good article.

The study fits within the topic of the Journal and is interesting for a potential reader. The introduction covers the basic knowledge on the field and authors of reference have been added.

The methods are well covered and they allow to fully understand and even replicate the study, which is something necessary in scientific papers.

However, I suggest the authors adding an explanation of the fitness centre characteristics. They have properly detected the limitation of focusing on a single fitness centers. For reducing the bias this can generate, I invite the authors to add some description of the center (business model, size, services offered, among others).

The results have been fully covered. The discussion is comprehensive, complete, and interesting for a potential reader.

Finally, I would like to add some minor comments:

  • Page 2, line 93. The mean age on this line is 27.87. However, this number does not completely match with the mean age of the participants on page 3, line109 nor on page 5, line 182. Please review this detail.
  • Likewise, the Standard Deviation of the mean age of the participants is stated to be 11.79 on page 5, line102. However, this number does not match with SD on page 2, line 93 nor with page 3, line 109.
  • Page 5, line 206: “The analysis of partial dependence plots (Figure 2)”. I am not sure that the authors refer to Figure 2 or Figure 1 instead. Please carefully review and modify, if applicable.
  • Figure 2 is quite difficult to read.

Lastly, I would like to congratulate again the authors for the good job done.

Author Response

We would like to thank the reviewer for his/her interest and time in assessing our manuscript. We also would like to thank his/her polite words for our work.

It was of great appreciation to receive the thoughtful comments and suggestions to improve the manuscript made by the reviewer.

Based on them we revised our manuscript in order to improve its content and quality.

Below we encompass the responses to the raised issues.

If there are any further changes or clarifications needed, please let us know. We will be happy to address them.

Comment: I suggest the authors adding an explanation of the fitness centre characteristics. They have properly detected the limitation of focusing on a single fitness centers. For reducing the bias this can generate, I invite the authors to add some description of the center (business model, size, services offered, among others).

Answer: We thank the reviewer for pointing this out, was added information about the fitness this was added to lines 92-95.

Comment: Page 2, line 93. The mean age on this line is 27.87. However, this number does not completely match with the mean age of the participants on page 3, line109 nor on page 5, line 182. Please review this detail.

    Likewise, the Standard Deviation of the mean age of the participants is stated to be 11.79 on page 5, line102. However, this number does not match with SD on page 2, line 93 nor with page 3, line 109.

Answer: We thank the reviewer for pointing this out, we confirmed the outputs of our analysis and was a typo error. Corrected line 110 and 189.

Comment: Page 5, line 206: “The analysis of partial dependence plots (Figure 2)”. I am not sure that the authors refer to Figure 2 or Figure 1 instead. Please carefully review and modify, if applicable.

Answer: We thank the reviewer for pointing this out, we corrected the error. Corrected line 200.

Comment: Figure 2 is quite difficult to read.

Answer: We thank the reviewer for pointing this out, we tried to correct this placing a  figure less complex explaining the overall idea.

Comment: The paper contains some sentences which do not read well making it dificult for the reader to understand what the author/s seek to communicate. The paper needs to be proofread preferably by an English proofreader.

Answer: The suggestion of the reviewer was taken into consideration and the manuscript had its English reviewed and properly proofread. We attached the editing certificate. However, if needed we can do another proofreading. 

Reviewer 3 Report

The study aims to evaluate the performance of machine learning algorithms in predicting user abandonment in fitness centres. The use of decision trees clearly shows all the information about it and can be very useful for fitness centres.

Congratulations for this research 

Improve quality and size of Figures 1 and 2 

Author Response

We would like to thank the reviewer for his/her interest and time in assessing our manuscript. It was of great appreciation to receive the thoughtful comments and suggestions to improve the manuscript made by the reviewer.

Comment: The study aims to evaluate the performance of machine learning algorithms in predicting user abandonment in fitness centres. The use of decision trees clearly shows all the information about it and can be very useful for fitness centres.

Answer: We thank the reviewer for his/her kind remark about our research.

Comment: Congratulations for this research. Improve quality and size of Figures 1 and 2

Answer: We improved the quality of figures 1 and 2. Additionally, figure two was corrected with a figure less complex explaining the overall idea

Reviewer 4 Report

I really enjoyed reading this interesting paper. I suggest thata native speaker checks the manuscript before it is published. Apart from that, I have a couple of minor points that should be addressed when revising the paper:

  1. the two paragraphs in line 76-80 and line 284-288 are virtually identical. Thus, one of them can be removed.
  2. Is something missing in lines 179 and 180? There are only two numbers (6. and 7.)
  3. the paragraph in lines 182-184 should either be placed ahead of or below Table 1
  4. Figure 2 is far too small and should be extended to a complete page

Author Response

We would like to thank the reviewer for his/her interest and time in assessing our manuscript. It was of great appreciation to receive the thoughtful comments and suggestions to improve the manuscript made by the reviewer. Based on them we revised our manuscript in order to improve its content and quality.

Below we encompass the responses to the raised issues.

If there are any further changes or clarifications needed, please let us know. We will be happy to address them.

Comment: I really enjoyed reading this interesting paper. I suggest thata native speaker checks the manuscript before it is published.

Answer: We thank the reviewer for his/her kind remark about our research. The suggestion of the reviewer was taken into consideration and the manuscript had its English reviewed and properly proofread. We attached the editing certificate. However, if needed we can do another proofreading.  

The two paragraphs in line 76-80 and line 284-288 are virtually identical. Thus, one of them can be removed.

Answer: Thank you very much for your kind comment and suggestion, the lines 282-286 was corrected to reduce its similarity with the sentence in lines 76-80. We understand the comment but if it’s possible to maintain in the discussion a similar idea allows us to reinforce this idea, which we think is important to future researches.

Is something missing in lines 179 and 180? There are only two numbers (6. and 7.)

 Answer: we acknowledge it, and this was corrected.

The paragraph in lines 182-184 should either be placed ahead of or below Table 1.

Answer: we acknowledge it, and we added it ahead of table 1.

Figure 2 is far too small and should be extended to a complete page

Answer: we acknowledge it, and the figure was corrected to a smaller one, to allow an easier interpretation and related to the text with the explanation. We hope that the reviewer agrees with this change.
